# *Pseudomonas aeruginosa* Community-Onset Bloodstream Infections: Characterization, Diagnostic Predictors, and Predictive Score Development—Results from the PRO-BAC Cohort

**DOI:** 10.3390/antibiotics11060707

**Published:** 2022-05-24

**Authors:** Pedro María Martínez Pérez-Crespo, Álvaro Rojas, Joaquín Felipe Lanz-García, Pilar Retamar-Gentil, José María Reguera-Iglesias, Olalla Lima-Rodríguez, Alfonso del Arco Jiménez, Jonathan Fernández Suárez, Alfredo Jover-Saenz, Josune Goikoetxea Aguirre, Eva León Jiménez, María Luisa Cantón-Bulnes, Pilar Ortega Lafont, Carlos Armiñanzas Castillo, Juan Sevilla Blanco, Jordi Cuquet Pedragosa, Lucía Boix-Palop, Berta Becerril Carral, Alberto Bahamonde-Carrasco, Teresa Marrodan Ciordia, Clara Natera Kindelán, Isabel María Reche Molina, Carmen Herrero Rodríguez, Inés Pérez Camacho, David Vinuesa García, Fátima Galán-Sánchez, Alejandro Smithson Amat, Esperanza Merino de Lucas, Antonio Sánchez-Porto, Marcos Guzmán García, Inmaculada López-Hernández, Jesús Rodríguez-Baño, Luis Eduardo López-Cortés

**Affiliations:** 1Infectious Diseases and Microbiology Unit, Hospital Universitario Virgen Macarena and Department of Medicine, University of Sevilla/Biomedicines Institute of Sevilla, 41009 Sevilla, Spain; pedrompc88@gmail.com (P.M.M.P.-C.); joaquinlanz@gmail.com (J.F.L.-G.); pilaretamar@hotmail.com (P.R.-G.); inlopezh@us.es (I.L.-H.); luiselopezcortes@gmail.com (L.E.L.-C.); 2Unidad de Enfermedades Infecciosas y Microbiología, Hospital Universitario Nuestra Señora de Valme, 41014 Sevilla, Spain; evaleon@ucpsevilla.es; 3Departamento de Enfermedades Infecciosas del Adulto, Facultad de Medicina, Pontificia Universidad Católica de Chile, Santiago 8331150, Chile; malvar77@yahoo.com; 4Servicio de Enfermedades Infecciosas, Hospital Regional Universitario de Málaga, IBIMA Málaga, 29010 Málaga, Spain; jmreguera99@yahoo.com; 5Unidad de Enfermedades Infecciosas, Servicio de Medicina Interna, Hospital Álvaro Cunqueiro, 36312 Vigo, Spain; olalla.lima@gmail.com; 6Grupo Enfermedades Infecciosas, Servicio de Medicina Interna, Hospital Costa del Sol, 29603 Marbella, Spain; alfarco@gmail.com; 7Unidad de Microbiología, Instituto de Investigación Sanitaria del Principado de Asturias (ISPA), Hospital Universitario Central de Asturias, 33011 Oviedo, Spain; jofersua@hotmail.com; 8Unidad Funcional de Infecciones Nosocomiales, Hospital Arnau de Vilanova, 25198 Lérida, Spain; ajover.lleida.ics@gencat.cat; 9Unidad de Enfermedades Infecciosas, Hospital Universitario de Cruces, 48903 Bizkaia, Spain; anejosune.goikoetxeaagirre@osakidetza.eus; 10Unidad de Cuidados Intensivos, Hospital Universitario Virgen Macarena, 41009 Sevilla, Spain; luisabulnes@hotmail.com; 11Unidad de Gestión Clínica de Enfermedades Infecciosas, Hospital Universitario de Burgos, 09006 Burgos, Spain; portegal@saludcastillayleon.es; 12Unidad de Enfermedades Infecciosas, Hospital Universitario Marqués de Valdecilla, Universidad de Cantabria, IDIVAL, 39008 Santander, Spain; carlos.arminanzas@scsalud.es; 13Unidad de Gestión Clínica de Enfermedades Infecciosas y Microbiología, Hospital Universitario Jerez de la Frontera, 11407 Jerez de la Frontera, Spain; juma27581@hotmail.com; 14Departamento de Medicina Interna, Hospital Universitario de Granollers, 08402 Granollers, Spain; jordicuquet@gmail.com; 15Unidad de Enfermedades Infecciosas, Hospital Universitari Mútua de Terrassa, 08221 Barcelona, Spain; luciaboix@hotmail.com; 16Unidad Clínica de Gestión de Enfermedades Infecciosas y Microbiología, Área Sanitaria del Campo de Gibraltar, 11207 Cádiz, Spain; mamorebece@gmail.com; 17Departamento de Medicina Interna, Hospital de El Bierzo, 24404 Ponferrada, Spain; med007783@me.com; 18Departamento de Microbiología Clínica, Complejo Asistencial Universitario de León (CAULE), 24071 León, Spain; tmarrodan@saludcastillayleon.es; 19Unidad Clínica de Enfermedades Infecciosas, Hospital Universitario Reina Sofía, 14004 Córdoba, Spain; clrntr@gmail.com; 20Unidad de Enfermedades Infecciosas, Hospital Universitario Torrecárdenas, 04009 Almería, Spain; irechemolina@gmail.com; 21Unidad de Gestión Clínica de Enfermedades Infecciosas y Microbiología Clínica, Complejo Hospitalario de Jaén, 23007 Jaén, Spain; gonees.data@hotmail.es; 22Unidad de Medicina Tropical, Hospital General de Poniente, 04700 El Ejido, Spain; inpercam@gmail.com; 23Unidad Gestión Clínica Enfermedades Infecciosas, Hospital Universitario Clínico San Cecilio, 18016 Granada, Spain; vinudav@yahoo.es; 24Unidad de Gestión Clínica de Microbiología, Hospital Universitario Puerta del Mar, 11009 Cádiz, Spain; fatima.galan@uca.es; 25Unidad de Medicina Interna, Fundació Hospital de l’Esperit Sant, 08923 Santa Coloma de Gramenet, Spain; asa30412@hotmail.com; 26Unidad de Enfermedades Infecciosas, Hospital Universitario General de Alicante, 03010 Alicante, Spain; merinoluc@gmail.com; 27Unidad de Enfermedades Infecciosas y Microbiología, Hospital de la Línea de la Concepción, 11300 La Línea de la Concepción, Spain; asanchezpo1314@gmail.com; 28Servicio de Medicina Interna, Hospital San Juan de la Cruz, 23400 Úbeda, Spain; m_guzman00@hotmail.es

**Keywords:** *Pseudomonas aeruginosa*, bacteraemia, bloodstream infection, community-onset, epidemiology

## Abstract

Community-onset bloodstream infections (CO-BSI) caused by gram-negative bacilli are common and associated with significant mortality; those caused by *Pseudomonas aeruginosa* are associated with worse prognosis and higher rates of inadequateempirical antibiotic treatment. The aims of this study were to describe the characteristics of patients with CO-BSI caused by *P. aeruginosa*, to identify predictors, and to develop a predictive score for *P. aeruginosa* CO-BSI. Materials/methods: PROBAC is a prospective cohort including patients >14 years with BSI from 26 Spanish hospitals between October 2016 and May 2017. Patients with monomicrobial *P. aeruginosa* CO-BSI and monomicrobial *Enterobacterales* CO-BSI were included. Variables of interest were collected. Independent predictors of *Pseudomonas aeruginosa* CO-BSI were identified by logistic regression and a prediction score was developed. Results: A total of 78patients with *P. aeruginosa* CO-BSI and 2572 with *Enterobacterales* CO-BSI were included. Patients with *P. aeruginosa* had a median age of 70 years (IQR 60–79), 68.8% were male, median Charlson score was 5 (IQR 3–7), and 30-daymortality was 18.5%. Multivariate analysis identified the following predictors of CO-BSI-PA [adjusted OR (95% CI)]: male gender [1.89 (1.14–3.12)], haematological malignancy [2.45 (1.20–4.99)], obstructive uropathy [2.86 (1.13–3.02)], source of infection other than urinary tract, biliary tract or intra-abdominal [6.69 (4.10–10.92)] and healthcare-associated BSI [1.85 (1.13–3.02)]. Anindex predictive of CO-BSI-PA was developed; scores ≥ 3.5 showed a negative predictive value of 89% and an area under the receiver operator curve (ROC) of 0.66. Conclusions: We did not find a good predictive score of *P. aeruginosa* CO-BSI due to its relatively low incidence in the overall population. Our model includes variables that are easy to collect in real clinical practice and could be useful to detect patients with very low risk of *P. aeruginosa* CO-BSI.

## 1. Introduction

Bloodstream infections (BSI) are a common consequence of invasive infections and cause significant morbidity and mortality worldwide. Their prognosis depends on several factors, including the aetiological agent, source of infection, appropriate antimicrobial treatment and the underlying conditions of the patient [1,2]. With respect to aetiological agents, invasive *Pseudomonas aeruginosa* infections are associated with high mortality rates, and prognosis is strongly associated with early active treatment, among other factors [3,4,5].

Since *P. aeruginosa* is predominantly a nosocomial pathogen, antipseudomonal antibiotics, especially antipseudomonal beta-lactams, in empirical regimens are often reserved for hospital-acquired infections to avoid overuse. Indeed, invasive infections due to *P. aeruginosa* outside the hospital setting are regarded as unusual [6,7]; in 2016, this microorganism caused 1.4% of community-acquired BSIs and 3.6% of healthcare-related BSIs in Spain [8]. However, inappropriate empirical antimicrobial treatment is associated with worse prognosis and mortality in *P. aeruginosa* BSI, including those that are community-onset [5]. Consequently, despite the low frequency, it would be useful to identify patients at risk of community-onset BSI due to *P. aeruginosa* in order to optimize their management.

The objectives of this study were to identify predictors of *P. aeruginosa* community-onset bloodstream infections (CO-BSI) and to develop a predictive score. In addition, we provide updated information on the epidemiological characteristics of community-onset *P. aeruginosa* bloodstream infections in Spain.

## 2. Material and Methods

### 2.1. Design, Study Sites and Periods

The PRO-BAC study was a multicentre prospective cohort study conducted in 26 Spanish hospitals (18 tertiary and 8 community hospitals) between 1 October 2016 and 31 March 2017, including all episodes of clinically significant BSI in adult patients (>14 years); no exclusion criteria were applied [8]. All methodological details were previously published [8]. Blood cultures were obtained, processed and interpreted in accordance with standard recommendations [9,10]. The study was approved by the Ethics Committees of the participating centres, which waived the need for informed consent due to the observational design, and it was registered in ClinicalTrials.gov (NCT03148769). The STROBE recommendations were followed in this study (Appendix A) [11].

For this analysis, episodes of monomicrobial CO-BSIin the PRO-BAC cohort caused by *P. aeruginosa* or any *Enterobacterales* were eligible. Predictors of *P. aeruginosa* CO-BSI were studied by comparing exposure to different variables with those of CO-BSI caused by *Enterobacterales*, the reason for this being that *P. aeruginosa* is often considered in patients when *Enterobacterales* are already a reasonable aetiological option.

### 2.2. Variables and Definitions

Variables collected include the type of hospital; admission ward; demographics; type and severity of chronic underlying diseases according to Charlson index [12]; acute severity of underlying disease according to Pitt score [13]; type of acquisition [14]; exposure to invasive procedures and devices in the previous month (including vascular catheter, urinary catheter, parenteral nutrition, mechanical ventilation and major surgery); antimicrobial use in the preceding month; aetiology of BSI; source of infection; presentation with severe sepsis or shock according to standard [15] and Sepsis-3 criteria [16]; antimicrobial treatment; and 30-day mortality. Data were collected according to an electronic case report form specifically designed by local investigators and supervised by Infectious Diseases and Critical Care physicians

BSIs were considered clinically significant if accompanied by signs or symptoms of infection, and contamination was ruled out [9,17]. In the case of potential contaminants, such as coagulase-negative staphylococci (CNS) or diphtheroids, only episodes in which the organism was isolated from ≥2 blood draws were considered [9].

Acquisition was considered community-onset if signs or symptoms of infection started <48 h after hospital admission or 48 h after hospital discharge, including strictly community-acquired cases and healthcare-related cases. Healthcare-related BSI was defined according to previously described criteria [14,18]. Source of BSI was based on clinical and laboratory data, using standard CDC criteria for secondary BSI [18]. For the diagnosis of catheter-related BSI, differential time to positivity was employed at all sites typically used for surgically implanted indwelling or difficult-to-replacecatheters. Subsequent episodes in the same patient caused by the same pathogen were excluded unless they occurred more than >3 months apart.

### 2.3. Statistical Analysis

Data are expressed as proportions for categorical variables, and as either means with standard deviation, or median with interquartile range (IQR) for continuous variables, as appropriate. Chi-square and Student’s t-tests were used to compare categorical and continuous variables, respectively. All time-dependent variables were measured with reference to the day when blood cultures were drawn (considered as day 0). All *p* values were two-tailed, and *p* values ≤ 0.05 were considered statistically significant. Independent predictors of CO-BSI due to *P. aeruginosa* bacteraemia were identified by logistic regression; variables with *p* value < 0.1 in univariate analysis and those considered potential predictors based on previous data were entered into the models, using a manual backward-stepwise selection procedure. Interactions between variables were checked. A predictive score was created using the final multivariate model; each predictive factor was assigned points, calculated by dividing the beta coefficients by the smallest beta coefficient in the model and rounding to the nearest unit. Positive predictive value (PPV), negative predictive value (NPV), sensitivity and specificity of the score were calculated. The predictive ability of the risk score for the observed data was checked by calculating the area under the receiver operating curve (AUROC). Data were analysed using SPSS version 23.0 (Chicago, IL, USA).

## 3. Results

### 3.1. Study Population

The PROBAC cohort included 5723 cases of monomicrobial bacteraemia, of which 3720 (65%) were classified as community-onset; 2680 (72%) of these were caused by gram-negative pathogens. *P. aeruginosa* caused 78 CO-BSI (1% of all CO-BSI and 3% of CO-BSI caused by gram-negatives); *Enterobacterales* caused 2572 CO-BSI (44.9% of all CO-BSI episodes, and 96% of those caused by gram-negatives).

Specific microorganisms isolated in the group of 2572 *Enterobacterales* CO-BSIs were *E. coli* (2064, 80.2%), *Klebsiella* spp. (265, 10.3%), *Proteus* spp. (74, 2.9%), *Enterobacter* spp. (49, 1.9%), *Salmonella* spp. (27, 1%), *Citrobacter* spp. (20, 0.8%), *Serratia marcescens* (16, 0.6%), *Morganella morganii* (12, 0.5%), and others (45, 1.8%).

### 3.2. Demographics, Comorbidities and Clinical Characteristics

Patient characteristics, sources of infection, clinical presentation, and CO-BSI mortality due to *P. aeruginosa* and *Enterobacterales* are shown in Table 1. Overall, patients with *P. aeruginosa* CO-BSI were younger, more frequently male, and had cancer, immunosuppressive conditions, healthcare-related BSI, and exposure to invasive procedures; the sources of BSI were also different (respiratory tract, unknown source and vascular catheter were more frequent among *P. aeruginosa* infections, while the urinary and biliary tract, and intraabdominal infections were less frequent). No statistically significant differences in severity were found at clinical presentation or ICU admission, although30-day mortality was higher among episodes caused by *P. aeruginosa* than than by *Enterobacterales*.

Specific antibiotic exposures in the 30 days prior to BSI are shown in Appendix A. Patients with *P. aeruginosa* had been more frequently exposed to antibiotics, and specifically to antipseudomonal beta-lactams.

### 3.3. Risk Factors and Predictive Score Development

A multivariate analysis using a logistic regression model showed that the independent risk factors for a *P. aeruginosa* aetiology in CO-BSI were male gender [1.89 (1.14–3.12), *p* = 0.01], haematological malignancy [2.45 (1.20–4.99), *p* = 0.01], obstructive uropathy [2.86 (1.13–3.02), *p* = 0.04], source of infection other than the urinary tract, biliary tract or intra-abdominal [6.69 (4.10–10.92), *p* < 0.01] and healthcare-associated BSI [1.85 (1.13–3.02), *p* = 0.01]. None of the antibiotic exposures shown in Appendix A were statistically significant risk factors after adjustment for other variables.

A predictive score was developed using the risk factors for *P. aeruginosa* CO-BSI identified in the multivariate analysis. The scores assigned to each significant variable according to their regression coefficient are listed in Table 2.The score had an area under the ROC curve of 0.66 (0.57–0.74), indicating a moderate predictive ability. The NPV, PPV, sensitivity and specificity for the different values of the score are shown in Table 3.

We also explored a model excluding patients with oncohaematological malignancies and neutropenia, as these patients would probably be candidates for antipseudomonal drugs anyway. The AUROC curve for the resulting model was 0.67 (0.55–0.74), showing a similar predictive ability.

### 3.4. Stratified Analysis by Source of Infection

To explorethe risk factors associated with *P. aeruginosa* CO-BSI versus *Enterobacterales* CO-BSI indifferent types of infection, a stratified analysis by source of bacteraemia was performed.

Among patients with urinary tract BSI, male sex [RR 1.55 (1.17–2.07), *p* = 0.03], cerebrovascular disease [RR 2.41 (1.26–4.61), *p* = 0.01], obstructive uropathy [RR 3.19 (1.84–5.51), *p* < 0.01], urinary catheter [RR 3.04 (1.76–2.25), *p* < 0.01], percutaneous nephrostomy [RR 3.65 (1.22–10.98), *p* = 0.02], ureteral stent [RR 3.81 (1.27–11.47), *p* = 0.01], transurethral resection [RR 8.35 (2–34.9), *p* < 0.01], healthcare-related acquisition [RR 1.68 (1.2–2.36), *p* = 0.02], cephalosporin use in the previous 30 days [RR 3.14 (1.40–7.08), *p* = 0.01] and carbapenem use in the previous 30 days [RR 5.32 (1.32–21.44), *p* = 0.01] were associated with an increased risk of *P. aeruginosa* CO-BSI.

Among those with unknown source of BSI, male sex [RR 1.55 (1.22–1.98), *p* = 0.04] and healthcare-related acquisition [RR 1.8 (1.3–2.48), *p* = 0.02] were associated with an increased risk of *P. aeruginosa* CO-BSI. Among biliary tract BSI patients, antibiotic use in the previous 30 days [RR 3.83 (2.31–6.33), *p* < 0.01] and biliary prosthesis [RR 4.12 (2.09–8.12), *p* < 0.01] were associated with an increased risk of *P. aeruginosa* CO-BSI. Obstructive biliary disease did not show an increased risk of *P. aeruginosa* CO-BSI compared to *Enterobacterales* CO-BSI [RR 1.32 (0.53–3.25), *p* = 0.26]. Intra-abdominal source of infection, oncological disease [RR 3.29 (2.64–4.1), *p* = 0.03], immunosuppressive treatment [RR 6.7 (4.7–9.49), *p* = 0.02], neutropenia <500 cells/µL [RR 25.86 (12.51–53.46), *p* < 0.01], antibiotic use in the previous 30 days [RR 5.32 (3.93–7.21), *p* = 0.03], use of indwelling vascular catheter in the previous week [RR 5.17 (3.84–6.96), *p* = 0.04], use of quinolones in the previous 30 days [RR 12.93 (2.7–61.8) *p* < 0.01] and anti-pseudomonal beta-lactams in the previous 30 days [RR 12.93 (2.7–61.8) *p* < 0.01] were all associated with *P. aeruginosa* CO-BSI.

## 4. Discussion

*P. aeruginosa* is an aetiology of concern in invasive infections as it is often associated with frail or immunocompromised patients and is associated with high mortality rates. Importantly, therapeutic options are more limited than for other common bacteria, and resistance is increasing worldwide [6,19,20,21,22]; in fact, the World Health Organization has listed carbapenem-resistant *P. aeruginosa* as a critical priority pathogen [23].

*P. aeruginosa* is considered an infrequent pathogen in non-nosocomial infections, despite the increasingly blurred boundaries between the hospital environment and the community. Furthermore, previous reports have shown that the epidemiological differences between community-acquired and healthcare-related BSI in all-cause bacteraemia were not well-define din the case of *P. aeruginosa* alone [24]. Treatment of *P. aeruginosa* is limited to a small number of antibiotics. Due to the well-known ability of *P. aeruginosa* to develop resistance, even during antibiotic treatment [20,21,25], empirical antipseudomonal antibiotics are mostly reserved for nosocomial infections. At the same time, delay in the administration of appropriate treatment is associated with increased mortality in *P. aeruginosa* BSI [3,4,5,26], which is crucial, especially when resistant or multi-drug-resistant *P. aeruginosa* is involved or suspected, due to the limited number of potentially useful antibiotics for these infections. However, the association between phenotypic resistance, virulence factors and biofilm-forming ability remains controversial [27].Thus, although the prevalence of *P. aeruginosa* as a cause of CO-BSI is low [7,8], it is necessary to identify in advance which patients would benefit from the use of antipseudomonal agents.

In this prospective multicentre bacteraemia cohort, we identified a substantial number of cases of *P. aeruginosa* CO-BSI and compared them with cases presenting with *Enterobacterales* CO-BSI. In our study, as reported previously [5], male sex, oncological disease, chronic pulmonary disease, haematological malignancy, immunosuppressive treatment and healthcare-related acquisition were more frequent in the *P. aeruginosa* CO-BSI group. In contrast toother studies [26,28], we observed that there were no significant differences in age or Charlson comorbidity index between *P. aeruginosa* CO-BSI and *Enterobacterales* CO-BSI after adjusting forsex, comorbidities and invasive procedures. Obstructive uropathy was also found to be an independent risk factor for *P. aeruginosa* CO-BSI. In the same vein, Esparcia et al. [29] developed a study only on community-onset urinary sepsis in the elderly, in which male sex, urinary tract neoplasia, recurrent urinary tract infection, indwelling urinary catheter and healthcare-related acquisition were found to be independent risk factors for community-onset *P. aeruginosa*.

To our knowledge, two studies reporting predictive scores for *P. aeruginosa* BSI have been published: the first one for all *P. aeruginosa* BSI cases, and the second for community-onset cases alone. Gransden et al. [30] presented a predictive score for all *P. aeruginosa* BSI. Eight variables were included; positive predictors were male gender, neutrophil count ≤1 × 10^9^ cells/L, previous/current antibiotic treatment, corticosteroid or cytotoxic treatment, hospital acquisition, intensive care unit patient, high-risk source of bacteraemia, and low-risk source of bacteraemia as a negative predictor. With a score > 3, the probability of BSI due to *P. aeruginosa* was 19.3%. Schechner et al. [7] described a prediction score for community-onset cases of *P. aeruginosa* BSI, which included the following variables: age > 90 years, recent antimicrobial use, presence of a urinary device and presence of a central venous catheter. When≥ 2 predictive factors were present, the probability of community-onset *P. aeruginosa* BSI was nearly 33%, and the area under the receiving operator curve of the multivariate model was0.726. Consistent with both prediction scores, our model has a low positive predictive value and a high negative predictive value for community-onset *P. aeruginosa* BSI, and a diagnostic accuracy comparable to that reported by Schechner et al. The main strengths of our model are that it includes variables that are easy to collect in real clinical practice and at the patient’s first contact with healthcare, allowing us to detect patients at very low risk of *P. aeruginosa* BSI.

Some limitations should be kept in mind. First, we did not collect data on the blood culture utilization rate, and although the participating sites collected the majority of blood cultures from their catchment population, we were unable to accurately calculate population-based incidence rates. Second, given the observational nature of our study, causality between the variables included in the model and community-onset *P. aeruginosa* BSI cannot be fully guaranteed. Nonetheless, similar variables have been previously explored for *P. aeruginosa* infections [5,7,30]. Furthermore, the low positive probability for community-onset *P. aeruginosa* BSI in our model, as in previously published models, implies the involvement of other unmeasured variables in the development of this disease. Future research should focus on identifying these as-yet undetected variables. Third, despite the fact that a large national prospective cohort was used, *P. aeruginosa* CO-BSI was uncommon and the frequency of cases was low. The limited predictive ability of the model is a likely corollary of this. Moreover, it has not been validated in an external cohort. Finally, and perhaps most importantly, our study inferred risk factors for pseudomonal aetiology in CO-BSI compared to *Enterobacterales*, limiting the population to Gram-negative bacteraemia. Because of this, the clinical utility of the model may be limited to the infections most frequently caused by gram negatives, such as those of biliary tract, intra-abdominal or urinary tract origin. The strengths of the study are prospective data collection and the high number of total cases recordedat26 hospitals indifferent regions of Spain with universal public healthcare coverage of the population, which provides a representative sample of *P. aeruginosa* CO-BSI in our country and makes it more easily generalizable.

## 5. Conclusions

*Pseudomonas aeruginosa* CO-BSI is a difficult entity to predict due to its relatively low incidence in the overall population. However, inappropriate empirical antimicrobial therapy is associated with a worse prognosis and higher mortality in *P. aeruginosa* CO-BSI. Our model could be useful to detect patients at very low risk of *P. aeruginosa* CO-BSI in cases of suspected Gram-negative bacteraemia, which would help to avoid overuse of antipseudomonal agents and thus reduce the selective pressure on this and other microorganisms.

## Figures and Tables

**Table 1 antibiotics-11-00707-t001:** Patient characteristics, comorbidities, exposure to invasive procedures in the previous 30 days and prosthesis wearers. Data are numbers (%) of cases, except where specified.

Variables	*Enterobacterales* (*n* = 2572)	*P. aeruginosa* (*n* = 78)	*p*
Male gender	1274/2572 (49.9)	53/78 (68.8)	<0.01
Age, median (IQR)	74 (62–83)	70 (60–79)	0.01
Charlson Index, median (IQR)	4 (3–6)	5 (3–7)	0.32
Diabetes mellitus	681/2572 (26.5)	16/78 (20.5)	0.24
Solid cancer	636/2572 (24.7)	29/78 (37.2)	0.01
Chronic renal insufficiency	353/2572 (13.7)	11/78 (14.1)	0.92
Dementia	294/2572 (11.4)	7/78 (9)	0.50
Chronic pulmonary disease	291/2572 (11.3)	14/78 (17.9)	0.07
Congestiveheartfailure	271/2572 (10.5)	8/78 (10.3)	0.94
Recurrent urinary tract infections	270/2572 (10.5)	5/78 (6.4)	0.24
Cerebrovascular disease	266/2572 (10.3)	10/78 (12.8)	0.48
Immunosuppressive treatment	233/2572 (9.1)	17/78 (21.8)	<0.01
Peripheral vascular disease	198/2572 (7.7)	5/78 (6.4)	0.67
Obstructive urinary disease	186/2572 (7.2)	11/78 (14.1)	0.02
Ischaemic cardiomyopathy	184/2572 (7.2)	3/78 (3.8)	0.26
Chronic hepatic insufficiency	180/2572 (7)	7/78 (9)	0.50
Obstructive biliary disease	154/2572 (6)	4/78 (5.1)	0.75
Haematological malignancy	91/2572 (3.5)	12/78 (15.4)	<0.01
Connective tissue disease	76/2572 (3)	2/78 (2.6)	0.84
Neutropenia < 500 cells/µL	36/2572 (1.4)	7/78 (9)	<0.01
Acquired immune deficiency syndrome	14/2572 (0.5)	2/78 (2.6)	0.02
**Healthcare-related factors**			
Healthcare-related acquisition	950 (36.9)	48/78 (61.5)	<0.01
Admission toacute care hospitalin the previous 60 days	432/2572 (16.8)	28/78 (35.9)	<0.01
Intravenous therapy ^a^	324/2572 (12.6)	32/78 (41)	<0.01
Outpatient care ^b^	313/2572 (12.2)	22/78 (28.2)	<0.01
Nursing home or long-term care facility ^a^	181/2572 (7)	2/78 (2.6)	0.13
Radiotherapy or chemotherapy ^a^	166/2572 (6.5)	20/78 (25.6)	<0.01
Woundcare or specialised nursing care at home ^a^	76/2572 (3)	5/78 (6.4)	0.08
Admission tochronic care hospital in previous 60 days	64/2572 (2.5)	2/78 (2.6)	0.097
Haemodialysis or peritoneal dialysis ^a^	43/2572 (1.7)	4/78 (5.1)	0.03
**Exposure toinvasive procedures in previous 30 days**			
Any vascular catheter	823/2572 (32)	37/78 (47.4)	<0.01
Previous antimicrobials	610/2572 (23.7)	29/78 (37.2)	<0.01
Urinary catheter	232/2572 (9)	13/78 (16.7)	0.02
Long-term vascular catheter ^c^	172/2572 (6.7)	19/78 (24.4)	<0.01
Major surgery	113/2572 (4.4)	2/78 (2.6)	0.44
Bronchoscopy	49/2572 (1.9)	4/78 (5.1)	0.05
Cystoscopy	16/2572 (0.6)	1/78 (1.3)	0.47
Mechanical ventilation	14/2572 (0.5)	0/78	0.51
Transurethral prostate resection	14/2572 (0.5)	2/78 (2.6)	0.02
Colonoscopy	13/2572 (0.5)	0/78	0.53
Gastroscopy	12/2572 (0.5)	0/78	0.55
Parenteral nutrition	9/2572 (0.3)	1/78 (1.3)	0.19
**Prosthesis wearers**			
Jointprosthesis	97/2572 (3.8)	0/78	0.08
Biliary prosthesis	82/2572 (3.2)	4/78 (5.1)	0.34
Ureteric stent	55/2572 (2.1)	3/78 (3.8)	0.31
Nephrostomy	52/2572 (2)	3/78 (3.8)	0.27
Pacemaker/Implantable cardioverter-defibrillator	48/2572 (1.9)	0/78	0.22
Prosthetic valves	34/2572 (1.3)	1/78 (1.3)	0.98
Osteosynthesis implant	24/2572 (0.9)	1/78 (1.3)	0.75
Vascular prosthesis	18/2572 (0.7)	0/78	0.46
Ventricular shuntcatheter	8/2572 (0.3)	0/78	0.62
**Source of bloodstream infection**			
Urinary tract	1520/2572 (49.1)	26/78 (33.3)	<0.01
Biliary tract	498/2572 (19.4)	7/78 (9)	0.02
Intraabdominal, non-biliary	181/2572 (7)	2/78 (2.6)	0.13
Unknown	176/2572 (6.8)	10/78 (12.8)	0.04
Respiratory tract	79/2572 (3.1)	13/78 (16.7)	<0.01
Skin and soft tissues	31/2572 (1.2)	4/78 (5.1)	<0.01
Vascular catheter	21/2572 (0.8)	6/78 (7.7)	<0.01
Osteoarticular	2/2572 (0.1)	1/78 (1.3)	<0.01
Endocarditis	2/2572 (0.1)	4/78 (5.1)	<0.01
Others	16/2572 (0.6)	2/78 (2.6)	0.04
**Severity and outcome**			
Pitt score > 3	148/2572 (5.8)	6/78 (7.7)	0.47
SOFA ≥ 2	785/2572 (30.5)	23/78 (29.5)	0.85
Severe sepsis/septic shock	678/2572 (26.4)	20/78 (25.6)	0.89
30-day mortality	285/2572 (10.9)	14/78 (17.9)	0.05

IQR: interquartile range; ^a^ Previous 30 days; ^b^ Two or more attendance to outpatient clinics; ^c^ Long-term vascular catheters includes: central venous catheters, Port-a-Cath, peripheral inserted central catheters and tunneled catheters.

**Table 2 antibiotics-11-00707-t002:** Independent risk factors for *Pseudomonas* spp. bloodstream infection in community-onset patients and assignment of scores based on regression coefficients obtained from multivariate logistic regression analysis. Dates expressed as adjusted OR ^a^ (95% CI), *p* value.

Variables	Adjusted ^a^ OR (95% CI)	*p*	Score Points
Male sex	1.89 (1.14–3.12)	0.01	1
Haematological malignancy	2.45 (1.20–4.99)	0.01	1
Obstructive uropathy	2.86 (1.13–3.02)	0.04	2
Source of infection other than urinary tract, biliary tract, or intra-abdominal	6.69 (4.10–10.92)	<0.01	4
Healthcare-related acquisition	1.85 (1.13–3.02)	0.01	1

^a^ ORs were adjusted forgender, age, Charlson index, comorbidities, exposure to invasive procedures and devices, source of infection and antimicrobial use in the preceding month.

**Table 3 antibiotics-11-00707-t003:** Proportion of patients, sensitivity, specificity, positive predictive value, and negative predictive values for different breakpoints according to the score predicting *Pseudomonas aeruginosa* bloodstream infection.

	Proportion of Patients	TP	FP	TN	FN	Sensitivity	Specificity	PPV	NPV	Accuracy
Score ≥ 0	100%	81	2572	0	0	1	0	0.03	-	0.03
Score ≥ 1	96.2%	73	2479	93	8	0.90	0.04	0.02	0.92	0.06
Score ≥ 2	88.5%	52	2297	275	29	0.64	0.11	0.02	0.90	0.12
Score ≥ 3	85.1%	39	2218	354	42	0.48	0.14	0.02	0.89	0.15
Score ≥ 4	84.2%	35	2200	372	46	0.43	0.14	0.02	0.89	0.15
Score ≥ 5	62%	28	1616	956	53	0.35	0.37	0.02	0.95	0.37
Score ≥ 6	19.3%	14	498	2074	67	0.17	0.81	0.03	0.97	0.79
Score ≥ 7	5.9%	6	150	2422	75	0.07	0.94	0.04	0.97	0.92
Score ≥ 8	1.2%	3	30	2542	78	0.04	0.99	0.09	0.97	0.96
Score ≥ 9	0%	0	0	2572	81	0	1	-	0.97	0.97

TP: True Positive; FP: False Positive; TN: True Negative; FN: False Negative; PPV: Positive Predictive Value; NPV: Negative Predictive Value.

## Data Availability

Not applicable.

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
