# Peer review of "Pseudomonas aeruginosa Community-Onset Bloodstream Infections: Characterization, Diagnostic Predictors, and Predictive Score Development—Results from the PRO-BAC Cohort"

_antibiotics, 2022, doi:10.3390/antibiotics11060707_

Round 1

Reviewer 1 Report

Dear authors 

1 - Introduction: it must be reformed in the content and in the writing of the general part to review the syntax of the topic   3- Discussion: to deepen in consideration of the problem of antibiotic resistance and the correlation between the ability to form biofilm, virulence factors and phonotypic resistance, use to deepen these studies: Learn more about this by using and citing the following references: PMID: 34572716 ; PMID: 35321081; PMID: 35453262    3 - Check the bibliographic entries throughout the text, some of which are non-compliant, review some entries in the bibliographic references .   4 - Review the English grammar and in particular the applied scientific English: in particular, the verb tenses and the syntax in the discussion.

Author Response

We thank the reviewer for these useful comments that helped us to correct and improve this paper. We made the suggested changes. We have made some changes to the Introduction and Discussion to try to emphasize the importance of the topic. We also have checked the bibliographic entries. The article is being revised by a native English-speaking colleague.

Reviewer 2 Report

This is an extensive study of bloodstream infections by Pseudomonas aeruginosa. It is an important research field of great importance in modern human medicine. It is a comprehensive article without unnecessary parts. I do not see any weaknesses that should hinder publication

Author Response

We really thank the reviewer for the gentle comments.

Reviewer 3 Report

Excellent submission, I do not have any comments concerning the methods, presentation and discussion of the data presented here.

Author Response

We thank the reviewer for the helpful comments.